# Zeta Hull Pursuits:
# Learning Nonconvex Data Hulls

**Yuanjun Xiong**[†]    **Wei Liu**[‡]    **Deli Zhao**[♯]    **Xiaoou Tang**[†]

[†]Information Engineering Department, The Chinese University of Hong Kong, Hong Kong
[‡]IBM T. J. Watson Research Center, Yorktown Heights, New York, USA
[♯]Advanced Algorithm Research Group, HTC, Beijing, China
{yjxiong,xtang}@ie.cuhk.edu.hk  weiliu@us.ibm.com  deli_zhao@htc.com

## Abstract

Selecting a small informative subset from a given dataset, also called column sampling, has drawn much attention in machine learning. For incorporating structured data information into column sampling, research efforts were devoted to the cases where data points are fitted with clusters, simplices, or general convex hulls. This paper aims to study nonconvex hull learning which has rarely been investigated in the literature. In order to learn data-adaptive nonconvex hulls, a novel approach is proposed based on a graph-theoretic measure that leverages graph cycles to characterize the structural complexities of input data points. Employing this measure, we present a greedy algorithmic framework, dubbed Zeta Hulls, to perform structured column sampling. The process of pursuing a Zeta hull involves the computation of matrix inverse. To accelerate the matrix inversion computation and reduce its space complexity as well, we exploit a low-rank approximation to the graph adjacency matrix by using an efficient anchor graph technique. Extensive experimental results show that data representation learned by Zeta Hulls can achieve state-of-the-art accuracy in text and image classification tasks.

## 1   Introduction

In the era of big data, a natural idea is to select a small subset of $m$ samples $\mathcal{C}_e = \{\boldsymbol{x}_{e_1}, \ldots, \boldsymbol{x}_{e_m}\}$ from a whole set of $n$ data points $\mathcal{X} = \{\boldsymbol{x}_1, \ldots, \boldsymbol{x}_n\}$ such that the selected points $\mathcal{C}_e$ can capture the underlying properties or structures of $\mathcal{X}$. Then machine learning and data mining algorithms can be carried out with $\mathcal{C}_e$ instead of $\mathcal{X}$, thereby leading to significant reductions in computational and space complexities. Let us write the matrix forms of $\mathcal{C}_e$ and $\mathcal{X}$ as $\mathbf{C} = [\boldsymbol{x}_{e_1}, \ldots, \boldsymbol{x}_{e_m}] \in \mathbb{R}^{d \times m}$ and $\mathbf{X} = [\boldsymbol{x}_1, \ldots, \boldsymbol{x}_n] \in \mathbb{R}^{d \times n}$, respectively. Here $d$ is the dimensions of input data points. In other words, $\mathbf{C}$ is a column subset selection of $\mathbf{X}$. The task of selecting $\mathbf{C}$ from $\mathbf{X}$ is also called by *column sampling* in the literature, and maintains importance in a variety of fields besides machine learning, such as signal processing, geoscience and remote sensing, and applied mathematics. This paper concentrates on solving the column sampling problem by means of graph-theoretic methods.

Existing methods in column sampling fall into two main categories according to their objectives: 1) approximate the data matrix $\mathbf{X}$, and 2) discover the underlying data structures. For machine learning methods using kernel or similar "N-Body" techniques, the Nyström matrix approximation is usually applied to approximate large matrices. Such circumstances include fast training of nonlinear kernel support vector machines (SVM) in the dual form [30], spectral clustering [8], manifold learning [25], *etc*. Minimizing a relative approximation error is typically harnessed as the objective of column sampling, by which the most intuitive solution is to perform uniform sampling [30]. Other non-uniform sampling schemes choose columns via various criteria, such as probabilistic samplings according to diagonal elements of a kernel matrix [7], reconstruction errors [15], determinant measurements [1], cluster centroids [33], and statistical leverage scores [21]. On the other hand, column sampling

may be cast into a combinatorial optimization problem, which can be tackled by using greedy strategies in polynomial time [4] and boosted by using advanced sampling strategies to further reduce the relative approximation error [14].

From another perspective, we are aware that data points may form some interesting structures. Understanding these structures has been proven beneficial to approximate or represent data inputs [11]. One of the most famous algorithms for dimensionality reduction, Non-negative Matrix Factorization (NMF) [16], learns a low-dimensional convex hull from data points through a convex relaxation [3]. This idea was extended to signal separation by pursuing a convex hull with a maximized volume [27] to enclose input data points. Assuming that vertices are equally distant, the problem of fitting a simplex with a maximized volume to data reduces to a simple greedy column selection procedure [26]. The simplex fitting approach demonstrated its success in face recognition tasks [32]. Parallel research in geoscience and remote sensing is also active, where the vertices of a convex hull are coined as endmembers or extreme points, leading to a classic "N-Finder" algorithm [31]. The above approaches tried to learn data structures that are usually characterized by convexity. Hence, they may fail to reveal the intrinsic data structures when the distributions of data points are diverse, *e.g.*, data being on manifolds or concave structures. Probabilistic models like Determinantal Point Process (DPP) [13] measure data densities, so they are likely to overcome the convexity issue. However, few previous work accessed structural information of possibly nonconvex data for column sampling/subset selection tasks.

This paper aims to address the issue of learning nonconvex structures of data in the case where the data distributions can be arbitrary. More specifically, we learn a nonconvex hull to encapsulate the data structure. The on-hull points tightly enclose the dataset but do not need to form a convex set. Thus, nonconvex hulls can be more adaptive to capture practically complex data structures. Akin to convex hull learning, our proposed approach also extracts extreme points from an input dataset. To complete this task, we start with exploring the property of graph cycles in a neighborhood graph built over the input data points. Using cycle-based measures to characterize data structures has been proven successful in clustering data of multiple types of distributions [34]. To induce a measure of structural complexities stemming from graph cycles, we introduce the *Zeta Function* which applies the integration of graph cycles to model the linkage properties of the neighborhood graph. The key advantage of the Zeta function is uniting both global and local connection properties of the graph. As such, we are able to learn a hull which encompasses almost all input data points but is not necessary to be convex. With structural complexities captured in the form of the Zeta function, we present a leave-one-out strategy to find the extreme points. The basic idea is that removing the on-hull points only has weak impact on structural complexities of the graph. The decision of removal will be based on *extremeness* of a data point. Our model, dubbed *Zeta Hulls*, is derived by computing and analyzing the extremeness of data points. The greedy pursuit of the Zeta Hull model requires the computation of the inversion of a matrix obtained from the graph affinity matrix, which is computationally prohibitive for massive-scale data. To accelerate such a matrix manipulation, we employ the *Anchor Graph* [18] technique in the sense that the original graph can be approximated with respect to the anchors originating from a randomly sampled data subset. Our model is testified through extensive experiments on toy data and real-world text and image datasets. Experimental results show that in terms of unsupervised data representation learning, the Zeta Hull based methods outperform the state-of-the-art methods used in convex hull learning, clustering, matrix factorization, and dictionary learning.

## 2  Nonconvex Hull Learning

To elaborate on our approach, we first introduce and define the *point extremeness*. It measures the degree of a data point being prone to lie on or near a nonconvex hull by virtue of a neighborhood graph drawn from an input dataset. As an intuitive criterion, the data point with strong connections in the graph should have the low point extremeness. To obtain the extremeness measure, we need to explore the underlying structure of the graph, where graph cycles are employed.

### 2.1  Zeta Function and Structural Complexity

We model graph cycles by means of a sum-product rule and then integrate them using a Zeta function. There are many variants of original *Riemann Zeta Function*, one of which is specialized in

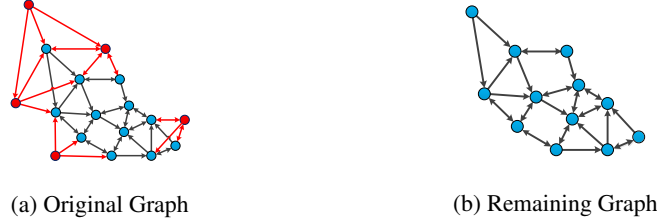

| (a) Original Graph | (b) Remaining Graph |

Figure 1: An illustration of pursuing on-hull points using the graph measure. (a) shows a point set with a $k$-nearest neighbor graph. Points in red are ones lying on the hull of the point set, *e.g.*, the points we tend to select by the Zeta Hull Pursuit algorithm. (b) shows the remaining point set and the graph after removing the on-hull points together with their corresponding edges. We observe that the removal of the on-hull (*i.e.*, "extreme") points yields little impact on the structural complexity of the graph.

weighted adjacency graphs. Applying the theoretical results of Zeta functions provides us a powerful tool for characterizing structural complexities of graphs. The numerical description of graph structures will play a critical role in column sampling/subset selection tasks.

Formally, given a graph $G(\mathcal{X}, E)$ with $n$ nodes being data points in $\mathcal{X} = \{\boldsymbol{x}_i\}_{i=1}^n$, let the $n \times n$ matrix $\mathbf{W}$ denote the weighted adjacency (or affinity) matrix of the graph $G$ built over the dataset $\mathcal{X}$. Usually the graph affinities are calculated with a proper distance metric. To be generic, we assume that $G$ is directed. Then an edge leaving from $\boldsymbol{x}_i$ to $\boldsymbol{x}_j$ is denoted as $e_{ij}$. A path of length $\ell$ from $\boldsymbol{x}_i$ to $\boldsymbol{x}_j$ is defined as $P(i, j, \ell) = \{e_{h_k t_k}\}_{k=1}^{\ell}$ with $h_1 = i$ and $t_\ell = j$. Note that the nodes in this path can be duplicate. A graph cycle, as a special case of paths of length $\ell$, is also defined as $\gamma_\ell = P(i, i, \ell)$ $(i = 1, \ldots, n)$. The sum-product path affinity $\nu_\ell$ for all $\ell$-length cycles can then be computed by $\nu_\ell = \sum_{\gamma_\ell \in \kappa_\ell} \nu_{\gamma_\ell} = \sum_{\gamma_\ell \in \kappa_\ell} w_{t_{\ell-1} h_1} \prod_{k=1}^{\ell-1} w_{h_k t_k}$, where $\kappa_\ell$ denotes the set of all possible cycles of length $\ell$ and $w_{h_k t_k}$ denotes the $(h_k, t_k)$-entry of $\mathbf{W}$, *i.e.*, the affinity from node $\boldsymbol{x}_{h_k}$ to node $\boldsymbol{x}_{t_k}$. The edge $e_{t_{\ell-1} h_1}$ is the last edge that closes the cycle. The computed compound affinity $\nu_\ell$ provides a measure for all cyclic connections of length $\ell$. Then we integrate such affinities for the cycles of lengths being from one to infinity to derive the graph Zeta function as follows,

$$\zeta_z(G) = \exp\left(\sum_{\ell=1}^{\infty} \nu_\ell \frac{z^\ell}{\ell}\right), \tag{1}$$

where $z$ is a constant. We only consider the situation where $z$ is real-valued. The Zeta function in Eq. (1) has been proven to enjoy a closed form. Its convergence is also guaranteed when $z < 1/\rho(\mathbf{W})$, where $\rho(\mathbf{W})$ is the spectral radius of $\mathbf{W}$. These lead to Theorem 1 [23].

**Theorem 1.** *Let $\mathbf{I}$ be the identity matrix and $\rho(\mathbf{W})$ be the spectral radius of the matrix $\mathbf{W}$, respectively. If $0 < z < 1/\rho(\mathbf{W})$, then $\zeta_z(G) = 1/\det(\mathbf{I} - z\mathbf{W})$.*

Note that $\mathbf{W}$ can be asymmetric, implying that $\lambda_i$ can be complex. In this case, Theorem 1 still holds. Theorem 1 indicates that the graph Zeta function we formulate in Eq. (1) provides a closed-form expression for describing the structural complexity of a graph. The next subsection will give the definition of the point extremeness by analyzing the structural complexity.

## 2.2 Zeta Hull Pursuits

From now on, for simplicity we use $\epsilon_G = \zeta_z(G)$ to represent the structural complexity of the original graph $G$. To measure the point extremeness numerically, we perform a leave-one-out strategy in the sense that each point in $\mathcal{C}$ is successively left out and the variation of $\epsilon_G$ is investigated. This is a natural way to pursue extreme points, because if a point $\boldsymbol{x}_j$ lies on the hull it has few communications with the other points. After removing this point and its corresponding edges, the reductive structural complexity of the remaining graph $G/\boldsymbol{x}_j$, which we denote as $\epsilon_{G/\boldsymbol{x}_j}$, will still be close to $\epsilon_G$. Hence, the point extremeness $\varepsilon_{\boldsymbol{x}_j}$ is modeled as the relative change of the structural complexity $\epsilon_G$, that is $\varepsilon_{\boldsymbol{x}_j} = \frac{\epsilon_G}{\epsilon_{G/\boldsymbol{x}_j}}$. Now we have the following theorem.

**Theorem 2.** *Given $\epsilon_G$ and $\epsilon_{G/\boldsymbol{x}_j}$ as in Theorem 1, the point extremeness measure $\varepsilon_{\boldsymbol{x}_j}$ of point $\boldsymbol{x}_j$ satisfies $\varepsilon_{\boldsymbol{x}_j} = (\mathbf{I} - z\mathbf{W})_{(jj)}^{-1}$, i.e., the point extremeness measure of point $\boldsymbol{x}_j$ is equal to the $j$-th diagonal entry of the matrix $(\mathbf{I} - z\mathbf{W})^{-1}$.*

**Algorithm 1** Zeta Hull Pursuits

---

**Input:** A dataset $\mathcal{X}$, the number $m$ of data points to be selected, and free parameters $z$, $\lambda$ and $k$.
**Output:** The hull of sampled columns $\mathcal{C}_e := \mathcal{C}_{m+1}$.
Initialize: construct $\mathbf{W}$, $\mathcal{C}_1 \leftarrow \varnothing$, $\mathcal{X}_1 = \mathcal{X}$, $\boldsymbol{c}_1 = \boldsymbol{0}$, and $\mathbf{W}_1 = \mathbf{W}$
**for** $i = 1$ **to** $m$ **do**
$\quad \varepsilon_{\boldsymbol{x}_j} := (\mathbf{I} - z\mathbf{W}_i)^{-1}_{(jj)}$, for $\boldsymbol{x}_j \in \mathcal{X}_i$
$\quad \boldsymbol{x}_{e_i} := \arg\min_{\boldsymbol{x}_j \in \mathcal{X}_i}(\varepsilon_{\boldsymbol{x}_j} + \frac{\lambda}{i}\boldsymbol{e}_j^\top \mathbf{W}\boldsymbol{c}_i)$
$\quad \mathcal{C}_{i+1} := \mathcal{C}_i \cup \boldsymbol{x}_{e_i}$
$\quad \boldsymbol{c}_{i+1} := \boldsymbol{c}_i + \boldsymbol{e}_{e_i}$
$\quad \mathcal{X}_{i+1} := \mathcal{C}_i / \boldsymbol{x}_{e_i}$
$\quad \mathbf{W}_{i+1} := \mathbf{W}_i$ with the $e_i$-th row and column removed
**end for**

---

According to previous analysis, the data point with a small $\varepsilon_{\boldsymbol{x}_j}$ tends to be on the hull and therefore has a strong extremeness. To seek the on-hull points, we need to select a subset of $m$ points $\mathcal{C}_e = \{\boldsymbol{x}_{e_1}, \ldots, \boldsymbol{x}_{e_m}\}$ from $\mathcal{X}$ such that they have the strongest point extremenesses. We formulate this goal into the following optimization problem:

$$\mathcal{C}_e = \arg\min_{\mathcal{C} \subset \mathcal{X}} \mathrm{g}(\mathcal{C}) + \lambda \boldsymbol{c}^\top \mathbf{W}\boldsymbol{c}, \tag{2}$$

where $\boldsymbol{c}$ is a *selection vector* with $m$ nonzero elements $c_{e_i} = 1$ $(i = 1, \ldots, m)$, and $\mathrm{g}(\mathcal{C})$ is the function which measures the impact on the structural complexity after removing the extracted points. In our case, $\mathrm{g}(\mathcal{C}) = \sum_{i=1}^m \varepsilon_{\boldsymbol{x}_{c_i}}$. The second term in Eq. (2) is a regularization term enforcing that the selected data points do not intersect with each other. It will enable the selection process to have a better representative capability. The parameter $\lambda$ controls the extent of the regularization.

Naively solving the combinatorial optimization problem in Eq. (2) requires exponential time. By adopting a greedy strategy, we can solve this optimization problem in an iterative manner and with a feasible time complexity. Specifically, in each iteration we extract one point from the current data set and add it to the subset of the selected points. Sticking to this greedy strategy, we will attain the desired $m$ on-hull points after $m$ iterations. In the $i$-th iteration, we extract the point $\boldsymbol{x}_{e_i}$ according to the criterion

$$\boldsymbol{x}_{e_i} = \arg\min_{\boldsymbol{x}_j \in \mathcal{X}_{i-1}} \varepsilon_{\boldsymbol{x}_j} + \frac{\lambda}{i}\boldsymbol{e}_j^\top \mathbf{W}\boldsymbol{c}_{i-1}, \tag{3}$$

where $\boldsymbol{e}_j$ is the $j$-th standard basis vector, and $\boldsymbol{c}_{i-1}$ is the selection vector according to $i-1$ selected points before the $i$-th iteration.

We name our algorithm *Zeta Hull Pursuits* in order to emphasize that we use the Zeta function to pursue the nonconvex data hull. Algorithm 1 summarizes the Zeta Hull Pursuits algorithm.

## 3 Zeta Hull Pursuits via Anchors

Algorithm 1 is applicable to small to medium-scale data $\mathcal{X}$ due to its cubical time complexity and quadratic space complexity with respect to the data size $|\mathcal{X}|$. Here we propose a scalable algorithm facilitated by a reasonable prior to tackle the nonconvex hull learning problem efficiently. The idea is to build a low-rank approximation to the graph adjacency matrix $\mathbf{W}$ with a small number of sampled data points, namely anchor points. We resort to the Anchor Graph technique [18], which has been successfully applied to handle large-scale hashing[20] and semi-supervised learning problems.

### 3.1 Anchor Graphs

The anchor graph framework is an elegant way to approximate neighborhood graphs. It first chooses a subset of $l$ anchor points $\mathcal{U} = \{\boldsymbol{u}_j\}_{j=1}^l$ from $\mathcal{X}$. Then for each data point in $\mathcal{X}$, its $s$ nearest anchors in $\mathcal{U}$ are sought, thereby forming an $s$-nearest anchor graph. The anchor graph theory assumes that the original graph affinity matrix $\mathbf{W}$ can be reconstructed from the anchor graph with a small number of anchors $(l \ll n)$. Anchor points can be selected by random sampling or a rough clustering process. Many algorithms are available to embed a data point to its $s$ nearest anchor points, as suggested in [18]. Here we adopt the simplest approach to build the anchor embedding matrix $\hat{\mathbf{H}}$; say, $\hat{h}_{ij} = \begin{cases} \exp\left(-d_{ij}^2/\sigma^2\right), & \boldsymbol{u}_j \in \{s \text{ nearest anchors of } \boldsymbol{x}_i\} \\ 0, & \text{otherwise} \end{cases}$, where $d_{ij}$ is the distance from data

---

**Algorithm 2**  Anchor-based Zeta Hull Pursuits

---
**Input:** A dataset $\mathcal{X}$, the number $m$ of data points to be sampled, the number $l$ of anchors, the number $s$ of nearest anchors, and a free parameter $z$.
**Output:** The hull of sampled columns $\mathcal{C}_e := \mathcal{C}_{m+1}$.
Initialize: construct $\mathbf{H}$, $\mathcal{X}_1 = \mathcal{X}$, $\mathcal{C}_1 = \varnothing$, and $\mathbf{H}_1 = \mathbf{H}$
**for** $i = 1$ **to** $m$ **do**
    perform SVD to obtain $\mathbf{H}_i := \mathbf{U}\mathbf{\Sigma}\mathbf{V}^T$
    $\varepsilon_{\boldsymbol{x}_j} := z \sum_{k=1}^{l} \frac{\lambda_j^2}{1-z\lambda_k^2}(U_{jk})^2$, for $\boldsymbol{x}_j \in \mathcal{X}_i$
    $\boldsymbol{x}_{e_i} := \arg\min_{\boldsymbol{x}_j \in \mathcal{X}_i}(\varepsilon_{\boldsymbol{x}_j} + \sum_{\boldsymbol{x}_t \in \mathcal{C}_i} \frac{\lambda}{i}\boldsymbol{h}_j\boldsymbol{h}_t^\top)$
    $\mathcal{C}_{i+1} := \mathcal{C}_i \cup \boldsymbol{x}_{e_i}$
    $\mathcal{X}_{i+1} := \mathcal{X}_i / \boldsymbol{x}_{e_i}$
    $\mathbf{H}_{i+1} := \mathbf{H}_i$ with the $e_i$-th row removed
**end for**

---

point $\boldsymbol{x}_i$ to anchor $\boldsymbol{u}_j$, and $\sigma$ is a parameter controlling the bandwidth of the exponential function. The matrix $\hat{\mathbf{H}}$ is then normalized so that its every row sums to one. In doing so, we can approximate the affinity matrix of the original graph as $\hat{\mathbf{W}} = \hat{\mathbf{H}}\mathbf{\Lambda}^{-1}\hat{\mathbf{H}}^\top$, where $\mathbf{\Lambda}$ is a diagonal matrix whose $i$-th diagonal element is equal to the sum of the $i$-th column of $\hat{\mathbf{H}}$. As a result, all matrix manipulations upon the original graph affinity matrix $\mathbf{W}$ can be approximated by substituting the anchor graph affinity matrix $\hat{\mathbf{W}}$ for $\mathbf{W}$.

### 3.2 Extremeness Computation via Anchors

Note that the computation of the point extremeness for $\varepsilon_{\boldsymbol{x}_j}$ depends on the diagonal elements of $(\mathbf{I} - z\mathbf{W})^{-1}$. Using the anchor graph technique, we can write $(\mathbf{I} - z\mathbf{W})^{-1} = (\mathbf{I} - z\mathbf{H}\mathbf{H}^\top)^{-1}$, where $\mathbf{H} = \hat{\mathbf{H}}\mathbf{\Lambda}^{-\frac{1}{2}}$. Thus we have the following theorem that enables an efficient computation of $\varepsilon_{\boldsymbol{x}_j}$. The proof is detailed in the supplementary material.

**Theorem 3.** *Let the singular vector decomposition of $\mathbf{H}$ be $\mathbf{H} = \mathbf{U}\mathbf{\Sigma}\mathbf{V}^\top$, where $\mathbf{\Sigma} = diag(\lambda_1, \ldots, \lambda_l)$. If $\mathbf{H}^\top\mathbf{H}$ is not singular, then $\varepsilon_{\boldsymbol{x}_j}^{-1} = 1 + z\sum_{k=1}^{l} \frac{\lambda_k^2}{1-z\lambda_k^2}(U_{jk})^2$, where $\mathbf{U} = \mathbf{H}\mathbf{V}\mathbf{\Sigma}^{-1}$ and $U_{jk}$ denotes the $(i,j)$-th entry of $\mathbf{U}$.*

Theorem 3 reveals that the major computation of $\varepsilon_{\boldsymbol{x}_j}$ will reduce to the eigendecomposition of a much smaller matrix $\mathbf{H}^\top\mathbf{H}$, which results in a direct acceleration of the Zeta hull pursuit process. At the same time, the second term of Eq. (3) encountered in the $i$-th iteration can be estimated by $\boldsymbol{e}_j^\top\mathbf{W}\boldsymbol{c}_i = \frac{1}{i}\sum_{\boldsymbol{x}_t \in \mathcal{C}_i} \boldsymbol{h}_j\boldsymbol{h}_t^\top$, where $\boldsymbol{h}_j$ denotes the $j$-th row of $\mathbf{H}$ and $\boldsymbol{c}_{i-1}$ is the selection vector of the extracted point set before the $i$-th iteration. These lead to the Anchor-based Zeta Hull Pursuits algorithm shown in Algorithm 2.

### 3.3 Downdating SVD

In Algorithm 2, the singular value decomposition dominates the total time cost. We notice that reusing information in previous iterations can save the computation time. The removal of one row from $\hat{\mathbf{H}}$ is equivalent to a rank-one modification to the original matrix. Downdating SVD [10] was proposed to handle this operation. Given the diagonal singular value matrix $\mathbf{\Sigma}_i$ and the point $\boldsymbol{x}_{e_i}$ chosen in the $i$-th iteration, the singular value matrix $\mathbf{\Sigma}_{i+1}$ for the next iteration can be calculated by the eigendecomposition of an $l \times l$ matrix $\mathbf{D}$ derived from $\mathbf{\Sigma}_i$, where $\mathbf{D} = (\mathbf{I} - \frac{1}{1+\mu}\boldsymbol{h}_{e_i}\boldsymbol{h}_{e_i}^\top)\mathbf{\Sigma}_i$, and $\mu^2 + \|\boldsymbol{h}_{e_i}\|_2^2 = 1$. The decomposition of $\mathbf{D}$ can be efficiently performed in $O(l^2)$ time [10]. Then the computation of $U_{i+1}$ is achieved by a multiplication of $U_i$ with an $l \times l$ matrix produced by the decomposition operation on $\mathbf{D}$, which permits a natural parallelism. Consequently, we can further accelerate Algorithm 2 by using a parallel computing scheme.

### 3.4 Complexity Analysis

We now analyze the complexities of Algorithms 1 and 2. For Algorithm 1, the most time-consuming step is to solve the matrix inverse of $n \times n$ size, which costs a time complexity of $O(n^3)$. The overall time complexity is thus $O(mn^3)$ for extracting $m$ points. In the implementation we can use

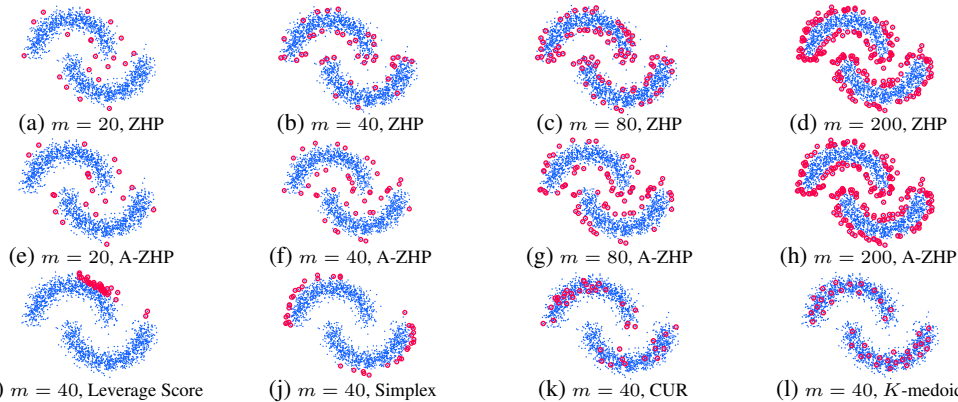

| (a) $m = 20$, ZHP | (b) $m = 40$, ZHP | (c) $m = 80$, ZHP | (d) $m = 200$, ZHP |
| (e) $m = 20$, A-ZHP | (f) $m = 40$, A-ZHP | (g) $m = 80$, A-ZHP | (h) $m = 200$, A-ZHP |
| (i) $m = 40$, Leverage Score | (j) $m = 40$, Simplex | (k) $m = 40$, CUR | (l) $m = 40$, $K$-medoids |

Figure 2: Zeta hull pursuits on the two-moon toy dataset. We select $m$ data points from the dataset with various methods. In the sub-figures, blue dots are data points. The selected samples are surrounded with red circles. The caption of each sub-figure describes the number of selected points $m$ and the method used to select those data points. First two rows shows the results of our algorithms with different $m$. The third row illustrates the comparisons with other methods when $m = 40$. For the leverage score approach, we follow the steps in [21].

the sparse matrix computation to reduce the constant factor [5]. For Algorithm 2, the most time-consuming step is to perform SVD over $\mathbf{H}$, so the overall time complexity is $O(mnl^2)$. Leveraging downdating SVD, we only need to calculate the full SVD of $\mathbf{H}$ once in $O(nl^2)$ time and iteratively update the decomposition in $O(l^2)$ time per iteration. The matrix multiplication operation then dominates the total time cost. Also, it can be parallelized using a multi-core CPU or a modern GPU, resulting in a very small constant factor in the time complexity. Since $l$ is usually less than $10\%$ of $n$, Algorithm 2 is orders of magnitude faster than Algorithm 1. For cases where $l$ needs to be relatively large ($20\%$ of $n$ for example), the computational cost will not show a considerable increase since $\mathbf{H}$ is usually a very sparse matrix.

## 4 Experiments

The Zeta Hull model aims at learning the structures of dataset. We evaluate how well our model achieves this goal by performing classification experiments. For simplicity, we abbreviate our algorithms as follows: the original Zeta Hull Pursuit algorithm (Algorithm 1), ZHP and its anchor version (Algorithm 2), A-ZHP. To compare with the state-of-the-art, we choose some renowned methods: $K$-medoids, CUR matrix factorization (CUR) [29], simplex volume maximization (Simplex) [26], sparse dictionary learning (DictLearn) [22] and convex non-negative matrix factorization (C-NMF) [6]. Basically, we use the extracted data points to learn a representation for each data point in an unsupervised manner. Classification is done by feeding the representation into a classifier. The representation will be built in two ways: 1) the sparse coding [22] and 2) the locality simplex coding [26]. To differentiate our algorithms from the original anchor graph framework, we conduct a set of experiments using the left singular vectors of the anchor embedding matrix $\mathbf{H}$ as the representation. In these experiments, anchors used in the anchor graph technique are randomly selected from the training set. To compare with existing low-dimension embedding approaches, we run the Large-Scale Manifold method [24] using the same number of landmarks as that of extracted points.

### 4.1 Toy Dataset

First we illustrate our algorithms on a toy dataset. The dataset, commonly known as "the two moons", consists of 2000 data points on the 2D plane which are manifold-structured and comprise nonconvex distributions. This experiment on the two moons provides illustrative results of our algorithms in the presence of nonconvexity. We select different numbers of column subsets $m = \{20, 40, 80, 200\}$ and compare with various other methods. A visualization of the results is shown in Figure 2. We can see that our algorithms can extract the nonconvex hull of the data cloud more accurately.

### 4.2 Text and Image Datasets

For the classification experiments in this section, we derive the two types of data representations (the sparse coding and the local simplex coding) from the points/columns extracted by compared meth-

Table 1: Classification error rates in percentage (%) on texts (TDT2 and Newsgroups) and hand-written number datasets (MNIST). The numbers in bold font highlight best results under the settings. In this table, "SC" refers to the results using the sparse coding to form the representation, while "LSC" refers to the results using local simplex coding. The cells with "-" indicate that the ZHP method is too expensive to be performed under the associated settings. The "Anchor Graph" refers to the additional experiments using the original anchor graph framework [18].

| Methods | TDT2 | | | | Newsgroups | | | | MNIST | | | |
|---|---|---|---|---|---|---|---|---|---|---|---|---|
| | $m = 500$ | | $m = 1000$ | | $m = 500$ | | $m = 1000$ | | $m = 500$ | | $m = 2000$ | |
| | SC | LSC | SC | LSC | SC | LSC | SC | LSC | SC | LSC | SC | LSC |
| **ZHP** | **2.31** | 1.97 | 1.53 | **0.48** | - | - | - | - | - | - | - | - |
| **A-ZHP** | 2.52 | 2.68 | 2.08 | 0.96 | 11.79 | 10.77 | **7.1** | **6.58** | 3.45 | **3.07** | 1.43 | **1.19** |
| Simplex [26] | 3.79 | **1.73** | 1.77 | 1.51 | 13.55 | **10.41** | 8.16 | 8.04 | 5.79 | 5.79 | 2.27 | 1.51 |
| DictLearn [22] | 3.73 | 5.62 | **1.18** | 2.57 | **9.51** | 10.76 | 6.72 | 9.63 | **3.16** | 3.16 | **1.36** | 2.11 |
| C-NMF [6] | 4.83 | 3.46 | 2.07 | 2.31 | 11.68 | 11.83 | 7.72 | 7.42 | 5.07 | 5.27 | 3.01 | 3.04 |
| CUR [29] | 6.82 | 3.73 | 2.37 | 1.52 | 15.32 | 11.44 | 12.38 | 9.47 | 10.13 | 10.13 | 3.79 | 5.27 |
| $K$-medoids [12] | 9.14 | 7.87 | 3.73 | 4.69 | 19.73 | 12.02 | 19.67 | 10.04 | 9.28 | 9.28 | 2.72 | 2.31 |
| Anchor Graph [18] | 5.81 | | 2.68 | | 12.32 | | 8.76 | | 3.17 | | 2.33 | |

Table 2: Recognition error rates in percentage (%) on object and face datasets. We select $L$ samples for each class in the training set for training or forming the gallery. The numbers in bold font highlight best results under the settings. In this table, "SC" refers to the results using the sparse coding to form the representation, while "LSC" refers to the results using local simplex coding. The "Raw Feature" refers to the experiments conducted on the raw features vectors. The face recognition process is described in Sec. (4.2).

| Methods | Caltech101 $d = 21504, L = 30$ | | | | Caltech101 $d = 5120, L = 30$ | | | | MultiPIE $d = 2000, L = 30$ | | | |
|---|---|---|---|---|---|---|---|---|---|---|---|---|
| | $m = 500$ | | $m = 1000$ | | $m = 500$ | | $m = 1000$ | | $m = 500$ | | $m = 2000$ | |
| | SC | LSC | SC | LSC | SC | LSC | SC | LSC | SC | LSC | SC | LSC |
| **A-ZHP** | **25.77** | 26.82 | **23.13** | 25.81 | 29.61 | **28.95** | **25.62** | **26.59** | 20.8 | **14.2** | 19.6 | **11.3** |
| Simplex [26] | 29.83 | **26.16** | 26.83 | **25.18** | 32.43 | 29.66 | 30.62 | 27.47 | 19.9 | 15.8 | **17.7** | 13.7 |
| DictLearn [22] | 26.95 | 29.73 | 26.73 | 29.51 | **29.15** | 31.83 | 28.93 | 29.67 | **19.6** | 20.8 | 18.5 | 19.7 |
| C-NMF [6] | 30.66 | 27.83 | 28.72 | 27.62 | 32.57 | 31.13 | 31.15 | 28.73 | 20.4 | 17.5 | 19.9 | 14.8 |
| CUR [21] | 29.74 | 28.77 | 26.16 | 26.81 | 31.69 | 32.57 | 30.72 | 31.13 | 21.3 | 21.9 | 20.7 | 21.6 |
| $K$-medoids [12] | 27.82 | 27.64 | 26.09 | 25.73 | 29.85 | 29.63 | 28.97 | 28.28 | 29.7 | 19.8 | 25.4 | 17.7 |
| Anchor Graph [18] | 26.32 | | 25.15 | | 30.53 | | 28.14 | | 17.6 | | 14.4 | |
| Large Manifold [24] | 28.71 | | 27.92 | | 32.67 | | 30.19 | | 31.4 | | 30.1 | |
| Raw Feature [28] | 26.7 | | | | 31.18 | | | | 27.6 | | | |

ods. By measuring the performance of applying these representations to solving the classification tasks, we can evaluate the representative power of the compared point/column selection methods.

*The sparse coding* is widely used for obtaining the representation for classification. Here a standard $\ell_1$-regularized projection algorithm (LASSO) [22] is adopted to learn the sparse representation from the extracted data points. LASSO will deliver a sparse coefficient vector, which is applied as the representation of the data point. We use "SC" to indicate the related results in Table 1 and Table 2.

*The local simplex coding* reconstructs one data point as a convex combination of a set of nearest exemplar points, which form local simplexes [26]. Imposing this convex reconstruction constraint leads to non-negative combination coefficients. The sparse coefficients vector will be used as data representation. "LSC" indicates the related results in Table 1 and Table 2.

*The classification pipeline* is as follows. After extracting $m$ points/columns from the training set, all data points will be represented with these selected points using the two approaches above. Then we feed the representations into a linear SVM for the training and testing. The better classification accuracy will reveal the stronger representative power of the column selection algorithm. In all experiments, the parameter $z$ is fixed at $0.05$ to guarantee the convergence of the Zeta function. We find that final results are robust to $z$ once the convergence is guaranteed. For the A-ZHP algorithm, the parameter $s$ is fixed at $10$ and the number of anchor points $l$ is set as $10\%$ of the training set size. The bandwidth parameter $\sigma$ of the exponential function is tuned on the training set to obtain a reasonable anchor embedding.

The classification of text contents relies on the informative representation of the plain words or sentences. Two text datasets are adopted for classification, i.e. the **TDT2** dataset and the **Newsgroups** dataset [2]. In experiments, a subset of TDT2 is used (TDT2-30). It has 9394 samples from 30 classes. Each feature vector is of 36771 dimensions and normalized into unit length. The training set contains 6000 samples randomly selected from the dataset and rest of the samples are used for

testing. The parameter $m$ is set to be $500$ and $1000$ on this dataset. The Newsgroups dataset contains $18846$ samples from $20$ classes. The training set contains $11314$, while the testing set has $7532$. The two sets are separated in advance [2] and ordered in time sequence to be more challenging for classifiers. The parameter $m$ is set to be $500$ and $1000$ on this dataset. The classification results are reported in Table 1.

For object and face recognition tasks we conduct experiments under three classic scenarios, the hand-written digits classification, the image recognition, and the human face recognition. Related experimental results are reported in Table 1 and Table 2.

The **MNIST** dataset serves as a standard benchmark for machine learning algorithms. It contains $10$ classes of images corresponding to hand-written numbers from $0$ to $9$. The training set has $60000$ images and the testing set has $10000$ images. Each sample is a $784$-dimensional vector.

The **Caltech101** dataset [17] is a widely used benchmark for object recognition systems. It consists of images from $102$ classes of objects ($101$ object classes and one background class). We randomly select $30$ labeled images from every class for training the classifier and $3000$ images for testing. The recognition rates averaged over all classes are reported. Every image is processed into a feature vector of $21504$ dimensions by the method in [28]. We also conduct experiment on a feature subset of the top $5000$ dimensions (Caltech101-5k). In this experiment, $m$ is set to be $500$ and $1000$. On-hull points are extracted on the training set.

The **MultiPIE** human face dataset is a widely applied benchmark for face recognition [9]. We follow a standard gallery-probe protocol of face recognition. The testing set is divided into the gallery set and the probe set. The identity predication of a probe image comes from its nearest neighbor of Euclidean distance in the gallery. We randomly select $30,000$ images of $200$ subjects as the training set for learning the data representation. Then we pick out $3000$ images of the other $100$ subjects ($L = 30$) to form the gallery set and $6000$ images as the probes. The head poses of all these faces are between $\pm 15$ degrees. Each face image is processed into a vector of $5000$ dimensions using the local binary pattern descriptor and PCA. We vary the parameter $m$ from $500$ to $2000$ to evaluate the influence of number of sampled points.

**Discussion.** For the experiments on these high-dimensional datasets, the methods based on the Zeta Hull model outperform most compared methods and also show promising performance improvements over raw data representation. When the number of extracted points grows, the resulting classification accuracy increases. This corroborates that the Zeta Hull model can effectively capture intrinsic structures of given datasets. More importantly, the discriminative information is preserved through learning these Zeta hulls. The representation yielded by the Zeta Hull model is sparse and of manageable dimensionality (500-2000), which substantially eases the workload of classifier training. This property is also favorable for tackling other large-scale learning problems. Due to the graph-theoretic measure that unifies the local and global connection properties of a graph, the Zeta Hull model leads to better data representation compared against existing graph-based embedding and manifold learning methods. For the comparison with the Large-Scale Manifold method [24] on the MultiPIE dataset, we find that even using 10K landmarks, its accuracy is still inferior to our methods relying on the Zeta Hull model. We also notice that noise may also affect the quality of Zeta hulls. This difficulty can be circumvented by running a number of well-established outlier removal methods such as [19].

## 5   Conclusion

In this paper, we proposed a geometric model, dubbed Zeta Hulls, for column sampling through learning nonconvex hulls of input data. The Zeta Hull model was built upon a novel graph-theoretic measure which quantifies the point extremeness to unify local and global connection properties of individual data point in an adjacency graph. By means of the Zeta function defined on the graph, the point extremeness measure amounts to the diagonal elements of a matrix related to the graph adjacency matrix. We also reduced the time and space complexities for computing a Zeta hull by incorporating an efficient anchor graph technique. A synthetic experiment first showed that the Zeta Hull model can detect appropriate hulls for non-convexly distributed data. The extensive real-world experiments conducted on benchmark text and image datasets further demonstrated the superiority of the Zeta Hull model over competing methods including convex hull learning, clustering, matrix factorization, and dictionary learning.

**Acknowledgement**   This research is partially supported by project #MMT-8115038 of the Shun Hing Institute of Advanced Engineering, The Chinese University of Hong Kong.

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
