[Supplementary Material]

# Zeta Hull Pursuits: Learning Nonconvex Data Hulls Supplementary Material

**Yuanjun Xiong**[†]    **Wei Liu**[‡]    **Deli Zhao**[♯]    **Xiaoou Tang**[†]

[†]Information Engineering Department, The Chinese University of Hong Kong, Hong Kong
[‡]IBM T. J. Watson Research Center, Yorktown Heights, New York, USA
[♯]Advanced Algorithm Research Group, HTC, Beijing, China
{yjxiong,xtang}@ie.cuhk.edu.hk  weiliu@us.ibm.com  deli_zhao@htc.com

## 1 Proof for Theorem 1

**Theorem 1.** *Let $\mathbf{I}$ be the identity matrix and $\rho(\mathbf{W})$ be the spectral radius of the matrix $\mathbf{W}$, respectively. If $0 < z < 1/\rho(\mathbf{W})$, then $\zeta_z(G) = 1/\det(\mathbf{I} - z\mathbf{W})$.*

*Proof.* By definition, $\nu_\ell$ and $\mathbf{W}$ are related as

$$\nu_\ell = \sum_{\gamma_\ell \in \kappa_\ell} \nu_{\gamma_\ell} = \text{tr}(\mathbf{W}^\ell), \tag{1}$$

where $\text{tr}(\mathbf{W}^\ell)$ denotes the trace of the matrix power $\mathbf{W}^\ell$. Suppose that the eigen-decomposition of $\mathbf{W}$ is $\mathbf{W} = \mathbf{Q}\mathbf{\Lambda}\mathbf{Q}^{-1}$, where the diagonal matrix $\mathbf{\Lambda} = diag(\lambda_1, \ldots, \lambda_n)$. Then we have

$$\zeta_z(G) = \exp(\text{tr}(\sum_{\ell=1}^{\infty} \frac{z^\ell}{\ell}\mathbf{W}^\ell)) = \exp(\text{tr}(\sum_{\ell=1}^{\infty} \frac{z^\ell}{\ell}\mathbf{Q}\mathbf{\Lambda}^\ell\mathbf{Q}^{-1}))$$

$$= \exp(\text{tr}(\sum_{\ell=1}^{\infty} \mathbf{\Lambda}^\ell)) \tag{2}$$

$$= \exp(\sum_{i=1}^{n}\sum_{\ell=1}^{\infty} \frac{1}{\ell}(z\lambda_i)^\ell), \tag{3}$$

where $\lambda_i$ is the $i$-th eigenvalue of $\mathbf{W}$. Recall that for $0 < x < 1$, $\ln(1-x) = \sum_{\ell=1}^{\infty} -\frac{x^\ell}{\ell}$. Since $|z\lambda_i| < 1$, we have

$$\zeta_z(G) = \exp(-\sum_{i=1}^{n} \ln(\mathbf{I} - z\lambda_i)) = 1/(\prod_{i=1}^{n}(\mathbf{I} - z\lambda_i))$$

$$= 1/\det(\mathbf{I} - z\Lambda) \tag{4}$$

$$= 1/\det(\mathbf{I} - z\mathbf{W}), \tag{5}$$

which completes the proof. □

## 2 Proof for Theorem 2

**Theorem 2.** *Given $\epsilon_G$ and $\epsilon_{G/\boldsymbol{x}_j}$ as in Theorem 1, the point extremeness measure $\varepsilon_{\boldsymbol{x}_j}$ of point $\boldsymbol{x}_j$ satisfies $\varepsilon_{\boldsymbol{x}_j} = (\mathbf{I} - z\mathbf{W})^{-1}_{(jj)}$, i.e., the point extremeness measure of point $\boldsymbol{x}_j$ is equal to the $j$-th diagonal entry of the matrix $(\mathbf{I} - z\mathbf{W})^{-1}$.*

*Proof.* By Theorem 1, the structural complexity of the remaining graph, $\epsilon_{G/\boldsymbol{x}_j}$, has the determinant form $\epsilon_{G/\boldsymbol{x}_j} = 1/\det(\mathbf{I} - z\mathbf{W}_{jj})$, where $\mathbf{W}_{jj}$ denotes the reduced matrix after removing the $j$-th column and $j$-th row of $\mathbf{W}$. Then we have

$$\varepsilon_{\boldsymbol{x}_j} = \frac{\det(\mathbf{I} - z\mathbf{W}_{jj})}{\det(\mathbf{I} - z\mathbf{W})}. \tag{6}$$

By definition of the adjugate matrix $\mathrm{adj}(\mathbf{I} - z\mathbf{W})$, we have

$$\mathrm{adj}(\mathbf{I} - z\mathbf{W})_{(jj)} = (-1)^{(j+j)}\det(\mathbf{I} - z\mathbf{W}_{jj}). \tag{7}$$

From the property of matrix inverse, we can write

$$(\mathbf{I} - z\mathbf{W})^{-1} = \frac{1}{\det(\mathbf{I} - z\mathbf{W})}\mathrm{adj}(\mathbf{I} - z\mathbf{W}). \tag{8}$$

Combining Eq. (6)(7)(8), we complete the proof. □

## 3  Proof for Theorem 3

**Theorem 3.** *Let the singular value decomposition of $\mathbf{H}$ be $\mathbf{H} = \mathbf{U}\boldsymbol{\Sigma}\mathbf{V}^{\top}$, where $\boldsymbol{\Sigma} = diag(\lambda_1, \ldots, \lambda_l)$. If $\mathbf{H}^{\top}\mathbf{H}$ is not singular, then $\varepsilon_{\boldsymbol{x}_j}^{-1} = 1 + z\sum_{k=1}^{l}\frac{\lambda_k^2}{1-z\lambda_k^2}(U_{jk})^2$, where $\mathbf{U} = \mathbf{H}\mathbf{V}\boldsymbol{\Sigma}^{-1}$ and $U_{jk}$ denotes the $(i,j)$-th entry of $\mathbf{U}$.*

*Proof.* The point extremeness measure is in the form

$$\varepsilon_{\boldsymbol{x}_j} = (\mathbf{I} - z\mathbf{H}\mathbf{H}^{\top})_{(jj)}^{-1}. \tag{9}$$

In Eq. (9), the left side can be expanded by the Woodbury identity [2]

$$(\mathbf{I} - z\mathbf{H}\mathbf{H}^{\top})^{-1} = \mathbf{I} + z\mathbf{H}(\mathbf{I} - z\mathbf{H}^{\top}\mathbf{H})^{-1}\mathbf{H}^{\top}. \tag{10}$$

Substituting $\mathbf{H} = \mathbf{U}\boldsymbol{\Sigma}\mathbf{V}^{\top}$ in Eq. (10) gives

$$\begin{aligned}(\mathbf{I} - z\mathbf{H}\mathbf{H}^{\top})^{-1} &= \mathbf{I} + z\mathbf{U}\boldsymbol{\Sigma}\mathbf{V}^{\top}(\mathbf{I} - z\mathbf{V}\boldsymbol{\Sigma}^2\mathbf{V}^{\top})^{-1}\mathbf{V}\boldsymbol{\Sigma}\mathbf{U}^{\top}\\ &= \mathbf{I} + z\mathbf{U}\boldsymbol{\Sigma}\mathbf{V}^{\top}\mathbf{V}(\mathbf{I} - z\boldsymbol{\Sigma}^2)^{-1}\mathbf{V}^{\top}\mathbf{V}\boldsymbol{\Sigma}\mathbf{U}^{\top}\\ &= \mathbf{I} + z\mathbf{U}\boldsymbol{\Sigma}(\mathbf{I} - z\boldsymbol{\Sigma}^2)^{-1}\boldsymbol{\Sigma}\mathbf{U}^{\top}.\end{aligned} \tag{11}$$

Note that $\boldsymbol{\Sigma}$ is a diagonal matrix. Expanding the right side of the identity above gives us

$$\varepsilon_{\boldsymbol{x}_j}^{-1} = (\mathbf{I} - z\mathbf{H}\mathbf{H}^{\top})_{(jj)}^{-1} \tag{12}$$

$$= 1 + z(\mathbf{U}\boldsymbol{\Sigma}^2(\mathbf{I} - z\boldsymbol{\Sigma}^2)^{-1}\mathbf{U}^{\top})_{(jj)} \tag{13}$$

$$= 1 + z\sum_{k=1}^{l}\frac{\lambda_k^2}{1 - z\lambda_k^2}(U_{jk})^2, \tag{14}$$

which completes the proof. □

## 4  Experiments

The performance of learning data representation on the Caltech dataset [1] is shown in Fig. 1. We illustrate the recognition rates when the number of labeled samples for training the classifier varies as $L = \{5, 10, 15, 20, 25, 30\}$ images per class.

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

Figure 1: The performance of learning data representation on Caltech101.We vary the number of labeled training samples per class as $L = \{5, 10, 15, 20, 25, 30\}$ to yield the recognition rates. The best representation scheme of each compared method when $L = 30$ is used for this figure.