[Reviews · NeurIPS 2014]

Submitted by Assigned_Reviewer_18

The paper presents a novel approach for column sampling when the data point clusters comprise of non-convex hulls. Column sampling is important in selecting a small subset of data that represents the properties of the original dataset. The presented approach is based upon the computation of Zeta hulls. The authors model the graph cycles by means of the sum-product rule and integrate them using the Zeta function. The authors set up the optimization problem as finding the subset of points with the strongest point extremenesses. The authors also present a scalable algorithm to solve the problem using the anchor graph method which allows the original weighted graph to be reconstructed from a small set of anchors. The authors then perform a low rank approximation of the anchor graph for scalability purpose. The paper also analyzes the time and space complexity.
The authors evaluate their methodology on synthetic as well as real data from text and image classification. They use several state of the art baseline methods to compare the performance of their model. Their results show that their method outperforms the baseline methods.
Summary: The paper provides a novel approach to perform column sampling using Zeta hull pursuits. The authors show promising performance improvements using several baseline methods on real data.

Submitted by Assigned_Reviewer_26

This paper presents a method for non-convex hull learning, with the goal of selecting a small set of representative instances from a dataset (column sampling). The core idea is to select points on the hull based on structural complexity scores of the graph computing using zeta functions. Since naive application of this process involves a matrix inversion, the authors also propose extension based on anchor graphs. The paper is fairly well written, and the proposed idea seems novel. Experimental results on multiple real datasets demonstrate effectiveness of the technique.

I wonder whether the authors have though have robustness of the algorithm to noise/outliers. It seems the algorithm will pick those up even though they are not ideal for column sampling. Secondly, I wonder how the proposed technique relates to DPPs (http://www.nowpublishers.com/articles/foundations-and-trends-in-machine-learning/MAL-044) which also focus on subset selection, although with diversity in mind. It will be very informative to learn about any relationship.

Summary: This paper presents a method for column sampling based on non-convex hull learning using zeta functions. The paper should be of interest to the NIPS community, although inclusion of relationship to other subset selection techniques such as DPP will strengthen the paper.

Submitted by Assigned_Reviewer_42

"Zeta Hull Pursuits: Learning Non-convex Data Hulls" addresses the problem of column sampling in datasets by finding points that do not alter significantly the Ihara zeta function of a graph (presumably constructed via a kernel). They provide establish some basic facts about the zeta function and the effect that removing a vertex has on it. These they use to provide a fast greedy algorithm for selecting 'outlying' vertices, and a faster version using anchors. They demonstrate on some standard datasets that using these points can improve classification accuracy.

The paper is fairly well written, with good organization and only minor typos. I have checked the proofs and did not find any mistakes. I am not sure that the method is motivated enough to merit acceptance, while it is in fact a novel idea.

The authors elude to the fact that this works for summarization of the data points without assuming that they are preserving the convex hull or enscribing ellipsoids. I believe that this is the primary merit of this paper, that they are able to find representative points in a combinatorial fashion. With that said, it is not exactly clear what is the motivation behind the use of the zeta function. If they had clearly argued for the use of points that minimize their epsilon objective, perhaps via the learning of non-convex hulls (which was never clearly defined) then my review might be better.

Also, I don't think that the Theorems are significant enough technical contributions to merit much attention. Theorem 1 reiterates an established fact regarding the a determinant form of the Zeta function (generalized to weighted graphs). Theorems 2 and 3 are straightforward results from matrix algebra and the SVD.

Summary: The paper is well written and organized and the method is novel. However, I do not believe that the use of the method is well defended and motivated by this paper.
Author Feedback
Author rebuttal: We thank the reviewers for their valuable comments. Here we address their main concerns.

Reviewer #18
We thank the reviewer for his/her very positive comments.

Reviewer #26
Q1. About the robustness of the algorithm to noise/outliers.
We thank the reviewer for raising this good question. Our column sampling algorithm is not completely immune to noise/outliers. However, a variety of methods, e.g., M-estimation, mean shift, mode seeking, are available for handling and removing noise/outliers, so we can choose an appropriate method for data preprocessing and then run our algorithm.

Besides, our Zeta hull learning model can directly detect noise/outliers. The basic idea is: the noise/outlier points usually distribute homogeneously, and meanwhile they appear quite different from inlier (normal) points. With our model, we found that the extremeness ranks of noise/outliers are very agglomerative, so we can identify them using a histogram of extremeness values, like the method presented in the following paper:

S. Byers and A.E. Raftery, "Nearest-neighbor clutter removal for estimating features in spatial point processes". Journal of The American Statistical Association, 93(442):577–584, 1998.

We didn’t include the content for noise/outliers removal due to the limited space. To make it clear, we will discuss the problem in the final version.

Q2. How the proposed technique relates to DPPs.
In our revised version, we will cite the DPP paper:

A. Kulesza and B. Taskar. "Determinantal Point Processes for Machine Learning."Foundations and Trends in Machine Learning 5.2-3 (2012): 123-286.

Although DPP and our proposed sampling technique can both deal with subset selection, they have several distinctions. DPP is a powerful probabilistic model, which allows a lot of applications other than subset selection. Our technique is a pure graph-theoretic approach which serves to select informative data points entirely based on the geometric structure, i.e., Zeta hull, underlying input data. While DPP and our approach both emphasize the diversity of the selected subset, our approach focuses more on the geometric informativeness of the selected subset. Such informativeness is measured by the structural complexity of input data points, which we use the Zeta function to capture. We will discuss the relationships with DPP in the final version to strengthen the paper.

Reviewer #42
Q1. The motivation behind the use of the Zeta function.
Previous methods such as volume maximization and convex hull learning assume a convex hull to enclose all data points. By using the Zeta function, we do not have to make the assumption of convexity. Applying the Zeta function to the weighted adjacency graph of data provides us a powerful tool for characterizing the structural complexity of the graph. The key advantage of using the Zeta function is uniting the global and local connection properties in the graph. Owing to this advantage, we can learn a hull which encompasses almost all data points but is not necessary to be convex.

The change of the Zeta function values when varying the subset delivers a measurement of the point extremeness. The basic rule is that removing an on-hull point should yield a small change to the epsilon value. We then formulate the rule as the optimization objective $\varepsilon$, and thus the point selected at each step will have the least $\varepsilon$ value.

Based on this rule, we can easily extract a representative subset from input data. These selected points together usually form a non-convex hull of input data, which turns out to have a strong representative power through extensive experiments on synthetic and real-world datasets.

Q2. The significance of theoretical results.
Theorem 1 guarantees that the Zeta function defined on a graph bears a closed-form expression, and this conclusion can be generalized to any weighted graph. This theoretical result enables us to investigate the structural complexity of weighted graphs via cycles which are capable of dealing with non-convex structures. However, few previous works utilized structural data information of non-convexity for column sampling/subset selection tasks.

Theorems 2 establishes a clear criterion for selecting on-hull points by formulating the selection rule as a tractable optimization objective. Theorem 3 provides a feasible direction for accelerating the selection process, reducing the time complexity from O(N^3) to O(Nml^2) (N is the dataset size, much larger than m and l). Theorems 2 and 3 directly lead to Algorithms 1 and 2. These theoretical results not only ensure the scalability of the proposed Zeta hull learning technique, but also justify the good performance that we observe in the experimental evaluations.

Therefore, we think that presenting these theoretical results in the paper is indispensable for readers' easy understanding of our technique.